# The Evaluation of the Four-Chamber Cardiac Dissection Method of the Fetal Heart as an Alternative to Conventional Inflow–Outflow Dissection in Small Gestational-Age Fetuses

**DOI:** 10.3390/diagnostics12010223

**Published:** 2022-01-17

**Authors:** Camelia Albu, Adelina Staicu, Roxana Popa-Stanilă, Cosmina Bondor, Bogdan Pop, Liviu Chiriac, Dan Gheban, Romeo Micu, Romulus Valeriu Flaviu Turcu, Simion Simon, Doinița Crișan, Florin Stamatian

**Affiliations:** 1Department of Pathology, “Iuliu Haţieganu” University of Medicine and Pharmacy, 400000 Cluj-Napoca, Romania; a.camelia82@yahoo.com (C.A.); pop.bogdan21@gmail.com (B.P.); dgheban@gmail.com (D.G.); doinitacrisan@gmail.com (D.C.); 2IMOGEN Centre of Advanced Research Studies, Emergency County Hospital, 400000 Cluj-Napoca, Romania; roxana.tania@gmail.com (R.P.-S.); florin_stamatian@yahoo.com (F.S.); 31st Department of Obstetrics and Gynecology, “Iuliu Haţieganu” University of Medicine and Pharmacy, 400000 Cluj-Napoca, Romania; romeomicu@hotmail.com; 4Department of Radiology, “Iuliu Haţieganu” University of Medicine and Pharmacy, 400000 Cluj-Napoca, Romania; 5Department of Medical Informatics and Biostatistics, “Iuliu Haţieganu” University of Medicine and Pharmacy, 400000 Cluj-Napoca, Romania; cosmina_ioana@yahoo.com; 6Department of Pathology, “Prof. Dr. Ion Chiricuta” Oncology Institute, 400015 Cluj-Napoca, Romania; 7Department of Medical Biophysics, “Iuliu Haţieganu” University of Medicine and Pharmacy, 400000 Cluj-Napoca, Romania; liviu_chiriac77@yahoo.ca; 8National Magnetic Resonance Centre, Babeș Bolyai University, 400000 Cluj-Napoca, Romania; flavius.turcu@phys.ubbcluj.ro (R.V.F.T.); simion.simon@phys.ubbcluj.ro (S.S.)

**Keywords:** magnetic resonance imaging, microscopy, macroscopy, heart defects, congenital, fetus diagnostic imaging, fetus pathology, high-field magnetic resonance imaging

## Abstract

The examination of very small fetal hearts requires special equipment and a specialist that are not available in many general pathology laboratories. Compared to conventional examination, the four-chamber cardiac dissection (4CCD) method can be performed by any pathologist using instruments generally available in pathology services. The aim of this study is to evaluate the efficiency of the 4CCD method in the examination of small fetal hearts using post-mortem magnetic resonance imaging (pm-MRI) at 7T as the standard. Twelve fetuses with gestational ages between 13 and 19 weeks have been included in this study. All fetuses underwent pm-MRI examination prior to pathologic examination. The 4CCD method was used for the cardiac examination in all cases following the same guidelines for cardiac sectioning. The 4CCD was able to identify all cardiac anatomic structures as compared to pm-MRI at 7T, demonstrating a sensibility of 95.8% (95% CI, 94.5–95.8) and specificity of 100% (95% CI, 32.3–100). The overall accuracy in identifying cardiac anatomic structures was 95.8% (95% CI, 93.4–95.8). Additionally, the 4CCD method was able to detect cardiac anomalies with an overall diagnostic accuracy of 91% (95% CI, 85.8–94.2), sensibility of 67.6% (95% CI, 54.5–75.3), and specificity of 97% (95% CI, 93.7–99) as compared to pm-MRI at 7T. The four-chamber view dissection method can be considered as an alternative to the conventional inflow–outflow dissection method in selected cases.

## 1. Introduction

Fetal cardiac examination is an important step in evaluating the presence of malformations during fetal autopsy and may present unique challenges. The combination of small-sized organ and complex malformative lesions require expertise, experience, and appropriate medical instruments in order to perform post-mortem examination of the fetal heart. Detailed clinical data and good communication with the obstetrician and cardiologist are crucial for a correct assessment of cardiac malformations [1].

A generally accepted method for cardiac dissection of the fetal heart is the inflow–outflow method. The sections for opening the atria and ventricles follow the course of blood flow through the heart. Another widely used method for cardiac dissection is the short axis method which is the election method for evaluating ischemic heart disease and is mostly used in children and adult patients [2].

The inflow–outflow cardiac dissection method can show any structural anomaly of the fetal heart and is the election method in evaluating fetal hearts by sequential segmental analysis [3]. However, when examining very small fetal hearts to perform the required sections for proper examination, microsurgical instruments and a stereomicroscope or at least a magnifying glass are required, along with experience in fetal cardiac examination [4]. Such a complex examination is time consuming and, in small pathology settings, practically impossible due to the lack of trained fetal pathology specialists and specific medical instruments [5]. Additionally, as described by Pacheco et al. in a study evaluating the pathologist’s effort in performing fetal, perinatal, and pediatric autopsies, a case with complex cardiac anomalies requires an increased amount of professional time compared to cases without cardiac anomalies [6]. The average time necessary to perform the fetal autopsies of fetuses below 20 weeks of gestation is 5 h [6,7]. Given the long time needed, the autopsies of small gestational-age fetuses represent a significant part of a pathologist’s workload.

Cardiac examination is a time-wise demanding procedure and represents a significant reason for increasing the examination time for small gestational-age fetuses.

Due to these facts, we considered introducing another cardiac dissection method, namely the 4CCD method, as an examination tool for small gestational-age fetuses [2].

To our best knowledge, the utility of the four-chamber view method for fetal cardiac examination was not evaluated until now. Although it was described previously as a dissection technique, its use was considered to bring less information compared to other dissection techniques [2].

Autopsy has been generally accepted as the gold standard when evaluating the accuracy of any diagnostic tool. To evaluate a new dissection method, it is required to have a different examination tool proven to give diagnostic results comparable to standard examination. The inflow–outflow method is the accepted standard pathologic examination of the fetal heart [4].

Since applying two different dissection methods on the same organ will not allow for proper assessment of the organ, we took into consideration post-mortem fetal cardiac imaging. 

Considering that conventional 1.5T and 3T magnetic resonance imaging (MRI) magnets currently used in clinical setting have shown lower accuracy in examining fetuses below 16 weeks of gestation [8,9], we considered the use of a high-power field magnet of 7T. Significant advances in the development of high-field MRI allowed for important progress in post-mortem fetal imaging [10,11] and numerous published studies have shown good diagnostic accuracy specially for fetal heart examination, describing pm-MRI as an alternative to conventional autopsy [12,13,14,15]. 

The proved diagnostic accuracy of 7T MRI in examining fetal hearts lead to the decision of using fetal pm-MRI examination at 7T as the standard to evaluate the efficiency of the post-mortem examination of small fetal hearts using the 4CCD method.

The aim was to evaluate if the 4CCD method can visualize standard cardiac structures compared to pm-MRI examination at 7T. 

Considering that the 4CCD method can examine only few transversal sections of the fetal heart, a second objective was to verify the ability of the 4CCD method to assess the malformative status of the evaluated cardiac structures compared to pm-MRI cardiac examination at 7T. 

## 2. Materials and Methods

### 2.1. General Considerations

Twelve consecutive second trimester fetuses with a gestational age ranging from 13 to 19 weeks of gestation (WG), calculated from the date of the last menstrual cycle, were included in this study. All fetuses resulted from therapeutic termination of pregnancy (TOP) due to plurimalformative syndromes or chromosomal anomalies carried out using prostaglandins administered locally and orally in 1st Clinic of Obstetrics and Gynecology Cluj-Napoca, Emergency County Clinical Hospital Cluj-Napoca, Romania, according to the internal protocol of the department. None of the pregnancies referred for therapeutic termination of pregnancy were for antepartum fetal death. 

After delivery, all cases were referred to IMOGEN—the Medical Research Institute within Emergency County Clinical Hospital Cluj-Napoca, Romania—between January 2015 and December 2017 for fetal autopsy and imagistic post-mortem examination.

The study protocol was approved by the “Iuliu Hatieganu” University of Medicine and Pharmacy Ethics Committee, Romania (no. 306/12 July 2017). Written informed parental consent for the scientific use of patient data was obtained before the imagistic and conventional autopsy procedures. All cases were handled according to the Human Tissue Act (2004).

Prior to pathological examination, all fetuses were immersed in 10% formalin solution at room temperature for 48 h to 1 week for proper fixation according to the internal protocol of the Pathology department of IMOGEN and stored in a refrigerator at 4° Celsius until transportation at the National Magnetic Resonance Centre for whole-body analysis using a 7 Tesla Bruker Biospec 70/16 USR (Bruker BioSpin MRI GmbH, Ettlingen, Germany) with a high magnetic field gradient unit (ball grid array 9 S HP), using a turbo spin-echo high-resolution T2-weighted imaging (T2 WI) protocol.

In the present study, we included fetuses without autolysis scanned at 7T with interpretable pm-MRI images of the heart and in which pathologic examination of the fetal heart was performed using the four-chamber view cardiac dissection method [2]. 

### 2.2. Post-mortem MRI Examination

The post-mortem fetal imaging scan at 7T required a medium time of 2 h and 30 min. The images were acquired in axial, sagittal, and coronal sections for 3 regions: the head and neck, thorax and abdomen, and pelvis. A 1H volume coil with an inner diameter of 60 mm was used. Details regarding fetuses’ positioning and the pilot protocol used for pm-MRI at 7T were described in previous research [15]. The sequence parameters are listed in Appendix A. After the scan, the fetuses were re-immersed in 10% formalin solution and transported the same day to the Pathology Department for autopsy.

### 2.3. Image Analysis

All images were analyzed by one radiologist with expertise in fetal and pediatric examination and by one embryologist with experience in embryo and fetal imaging blinded to the prenatal ultrasound results. The images were processed with the GE AW Workstation 4.6 (GE Healthcare, Chicago, IL, USA). Each specialist analyzed all images separately and the final diagnosis was approved by both specialists. The agreement index was not considered due to the different specialties of the imaging evaluators. 

According to Anderson and Shirali [3] in the sequential segmental analysis of the heart, for each case, the atrial, ventricular, and arterial segments and their connections; the atrioventricular valves; and the great vessels’ semilunar valves were assessed. 

Fourteen distinct anatomical structures were evaluated by both examination methods: caval veins, pulmonary veins, the right atrium, the left atrium, the interatrial septum, the tricuspid valve, the mitral valve, the right ventricle, the left ventricle, the interventricular septum, the infundibulum, the pulmonary artery, the aorta, and the arterial duct.

For each structure, the imagists examined normal or abnormal morphological aspects and mentioned the observed abnormal morphological aspects. Figure 1 depicts the fourteen cardiac items evaluated.

The data set included 168 structures (12 cases × 14 structures for each case). All items included were properly visualized using pm-MRI.

### 2.4. Pathologic Examination

Following the post-mortem MRI examination, all fetuses were submitted for pathologic examination. Cardiac examination using the 4CCD method included two steps: first, a macroscopic external examination of the fetal heart and, second, a four-chamber view dissection followed by microscopic examination of the four chamber view sections. External heart examination and assessment of great vessels were performed using an Olympus SZ61 stereomicroscope (Olympus Europe SE & Co, Hamburg, Germany) with standard specifications, a magnification from 0.67× to 4.5×, a zoom ratio of 6.7, a working distance of 110 mm, and a tube tilting angle of 45°, without auxiliary objectives. Macroscopic images were captured using a Canon EOS 1200D camera and EOS Digital Solution Software (Canon GmbH, Bucharest, Romania) for image acquisition. 

After external examination, two longitudinal sections were performed parallel with the diaphragmatic surface of the heart, as depicted in Figure 2a,b. The first section was performed through both ventricles and both atria, obtaining a four-chamber view. The second section was performed parallel to the first through the infundibulum and pulmonary artery valve. All obtained sections were processed using a vacuum infiltration processor, namely Tissue-Tek VIP 5 Jr (Sakura, Alphen aan den Rijn, The Netherlands), and embedded in paraffin blocks.

For very small fetal hearts weighting below 1 g, due to the very soft consistency of the cardiac tissue even after proper fixation, following macroscopic examination and prior to sectioning, the whole heart was submitted to the tissue processing protocol described previously, thus obtaining a firm consistency that allowed for proper sectioning of the fetal heart. The longitudinal sections followed the same landmarks we used for hearts with weights equal or above 1 g.

Three serial sections were obtained from each paraffin block and stained with hematoxylin-eosin. For the microscopic examination, we used an Olympus BX46 clinical microscope (Olympus Europe SE & Co, Hamburg, Germany) with an LED illuminator configured with a 2× plan apochromatic objective and dedicated image acquisition camera as well as software. 

To obtain panoramic views, representative slides were scanned using a slide scanner (3D Histech Pannoramic Scan, Budapest, Hungary) with 20× Plan-Apochromat objective. The images were acquired using dedicated digital slide viewer software (3D Histech Case Viewer 2.2, Budapest, Hungary). 

Due to inclusion and sectioning techniques, microscopic sections of the inferior side of the heart were a mirrored view of the macroscopy because the section was flipped over during the paraffin embedding. For the other two sections, the left–right sides corresponded to the external cardiac view, as depicted in Figure 2c–h and Figure 3, hence, after sectioning, they were included without flipping.

The pathologic four-chamber examination protocol is described schematically in Figure 4.

### 2.5. Statistical Analysis

For statistical analysis Microsoft Excel 2016^®^, the R package add-in for R Commander [16] and JavaStat for two-way contingency table analysis [17] were used. Numerical data was presented using arithmetic means ± standard deviation (SD) if the data was normally distributed or median (25–75th percentile) if not.

In order to evaluate the efficiency of the four-chamber view cardiac dissection method of small fetal hearts compared to pm-MRI at 7T, the Sensitivity (Se), Specificity (Sp), Positive Predictive Value (PPV), Negative Predictive Value (NPV), accuracy, and their 95% confidence intervals were calculated using the R package add-in for R Commander [16] and the online Interactive Statistics page [17]. Cohen’s kappa coefficient of agreement with the 95% confidence interval and the McNemar test were also used. The information was centralized in a database. For each cardiac structure, it’s presence and assessed of the malformative status were noted separately. All information was analyzed by a statistician without medical knowledge. 

The data supporting the findings described in this study are available from the corresponding author upon reasonable request.

## 3. Results

### 3.1. The Diagnostic Value of the Four-Chamber Examination Method to Identify Anatomic Cardiac Structures of Interest as Compared to Pm-MRI at 7T

The cases comprised in our group had a median gestational age (WG) of 16.5 weeks (min 15.25 g; max 17.25 g) and mean gestational age of 16.08 weeks ± 2.07SD, with a median fetal weight of 89 g (min 52.5 g; max 160.5 g) and mean fetal weight of 118.33 g ± 98.52SD, as well as a median fetal heart weight of 1 g (min 0.82 g; max 1.17 g) and mean fetal heart weight of 1.2 g ± 1.1SD. The detailed characteristics of all the cases included in our study, pm-MRI findings, and 4CCD findings, along with the complete autopsy findings, are presented in Table 1. 

In all the studied cases, 4CCD identified the superior and inferior vena cava, pulmonary veins, right and left atria, right and left ventricles, interventricular septum, aorta, and arterial duct.

The 4CCD failed to completely visualize four structures out of the 168 examined structures, representing 2.38% of all the examined structures, as follow: in one case of 17 WG with 1 g of the heart the interventricular septum; in one case of 13 WG and a fetal heart weight of 0.26 g, the interatrial septum; and in another 13 WG fetuses with 0.12 g of the heart, the interventricular septum and tricuspid valve. In all cases, the examination of these structures was compromised by oblique sectioning of the fetal heart. 

Pm-MRI at 7T confirmed the normal anatomy in the first two cases (case No. 5 and case No.10). In the third case, the incorrect macroscopic sectioning led to an incorrect diagnosis of the hypoplastic left ventricle (case No. 9). 

The 4CCD method has identified the anatomic cardiac structures of interest as compared to pm-MRI at 7T with an overall accuracy of 95.8% (95% CI, 93.4–95.8), demonstrating a Se of 95.8% (95% CI, 94.5–95.8) and Sp of 100% (95% CI, 32.3–100), as well as a PPV of 100% (95% CI, 98.7–100) and PNV of 30% (95% CI, 9.7–30). Additionally, the Cohen’s kappa coefficient of correlation was *k* = 0.45, revealing a moderate agreement between the two methods.

### 3.2. The Diagnostic Value of the Four-Chamber Examination Method to Assess the Malformative Status of the Fetal Cardiac Structures as Compared to Pm-MRI at 7T

When we assessed the malformative status of the examined cardiac structures, the overall diagnostic accuracy of 4CCD compared to pm-MRI at 7T (used as the golden standard) in detecting cardiac anomalies of the examined structures was 91% (95% CI, 85.8–94.2), with a Se of 67.6% (95% CI, 54.5–75.3), Sp of 97% (95% CI, 93.7–99), PPV of 85.2% (95% CI, 68.7–94.8), and NPV of 92.2% (95% CI, 89–94). Cohen’s kappa coefficient of agreement was *k* = 0.7, underlining an excellent concordance between the two methods.

Table 2 describes the overall statistical analysis regarding the diagnostic capacity of 4CCD compared to pm-MRI at 7 T for each cardiac structure considered for evaluation.

All hearts diagnosed as normal by pm-MRI at 7T were also diagnosed as normal by 4CCD. Most differences were related to the subjective interpretation of diameters and wall thickness, as follows: 

In case No.7, pm-MRI diagnosed pulmonary artery stenosis and arterial duct stenosis, while 4CCD interpreted the heart as normal. 

In case No.8, both methods visualized aortic stenosis but pm-MRI also diagnosed a left ventricle hypertrophy and dilation of the left atrium, which 4CCD failed to diagnose. In this case, 4CCD diagnosed a millimetric ventricular septal defect. 

In case No.11, both methods diagnosed the complete atrioventricular canal but pm-MRI also diagnosed a hypoplastic left ventricle as well as an overriding aorta and arterial duct agenesis. Comparative images with observed lesions are depicted in Figure 5. Additionally, in case No.12, though both methods described most of the lesions, 4CCD failed to describe bilateral ventricular hypertrophy and a ventricular septal defect. Thus, 4CCD overdiagnosed an aortic valve stenosis and tricuspid valve stenosis. Comparative images are presented in Figure 6. Therefore, as shown in Table 2, the lowest sensitivity rate was observed in examining the arterial duct and left atrium. The disagreement was in two out of three cases, presenting anomalies on pm-MRI (cases No. 7 and 11). For the conventional dissection method, a minimum of six fine sections are required for evaluation of the atria, ventricles, and outflow tracts, whereas for the 4CCD method, only two parallel sections are required. External cardiac examination was described as a common step for both methods. Microscopic examination required more time for the 4CCD method, especially for malformed hearts, but it did not surpass the entire time needed for the stereomicroscopic cardiac examination. Our subjective assessment revealed that the 4CCD method required less time for examining fetal hearts than the conventional inflow–outflow method for very small fetal hearts.

## 4. Discussion

The present study demonstrates that the 4CCD method can properly visualize standard cardiac structures and evaluate their malformative status in late first and second-trimester fetuses with an average weight of 89 g.

As expected, there was a difference between the efficiency markers, with the 4CCD method having a higher accuracy in detecting anatomic cardiac structures (95.8% (95% CI, 93.4–95.8)) and a lower accuracy of 91% (95% CI, 85.8–94.2) in assessing their malformative status. As opposed to pm-MRI, the 4CCD method allows for examination of only few cardiac sections and some small cardiac anomalies could be missed at microscopic examination. Additionally, as further presented, an incorrectly performed section could lead to misdiagnosis or incomplete diagnosis.

In our study, we observed overall good Se (95.8% (95% CI, 94.5–95.8)) and Sp (100% (95% CI, 32.3–100)) in identifying cardiac anatomic structures. The effectiveness of the method was increased due to the combination of macroscopic and microscopic examinations since the anatomic structures evaluated macroscopically had a Sensitivity and Specificity of 100%.

When assessing diagnostic accuracy, the lowest sensitivity rate of 25% (95% CI 0.63–80.59) was observed in examining the arterial duct. Even though it was one of the structures examined mostly macroscopically, due to a normally extremely small diameter of the arterial duct and lack of normal standards for the vascular diameter according to gestational age, macroscopic evaluation of a slightly dilated arterial duct (1.4 mm measured by pm-MRI) is difficult to assess. Other small sensitivity rates were observed in the evaluation of the right atrium, right ventricle, interventricular septum, left ventricle, and pulmonary artery. In all cases, the discordance was due either to difficulties in the pathologic evaluation of the diameter and thickness of the cardiac structures due to the lack of normal standards for gestational age, or misdiagnosis due to the oblique macroscopic sectioning of the fetal heart. In seven out of the 14 examined anatomic structures, the Se was above 80%.

Pm-MRI images at different levels give a useful overview of vascular and cardiac chambers’ diameters that can easily highlight a dilated or stenotic structure [18]. As opposed to the 4CCD method, it allows for a bidimensional limited number of sections which cannot always correctly assess a vascular diameter, especially if secondary changes in connected cardiac structures are absent or if the stenosis or dilation is not severe. 

A complete diagnostic agreement between pathology and pm-MRI was only of 33.33% in cases with a gestational age of 13 weeks and higher in cases with a gestational age above 16 weeks. This was due mostly to the fact that in the group with fetuses of 13 weeks of gestation, two out of the three cases had complex cardiac anomalies. Additionally, partial agreement was observed mostly in cases with either complex cardiac anomalies or with dilation or stenosis of diverse structures. We had no case of complete disagreement between the pathology and pm-MRI diagnosis in our study. 

Although the time needed for the microscopic examination of the cardiac sections was longer than the microscopic evaluation of the myocardium after conventional cardiac dissection, we empirically observed an overall shorter time for the cardiac evaluation in very small fetal hearts. Additionally, the fetal cardiac examination using the 4CCD method retrieved good results, did not require any special equipment, and was performed using standard dissection equipment found in any pathology service.

Compared to the standard inflow–outflow dissection method, which is considered to be the gold standard for fetal cardiac examination as it can evidentiate any structural cardiac anomaly, the 4CCD method had an accuracy of 91% (95% CI, 85.8–94.2) in detecting cardiac anomalies. We consider that conventional dissection should be used whenever possible. However, when the examination of small fetal hearts by conventional dissection is difficult due to maceration or increased tissular friability, or impossible due to the absence of specific medical instruments and trained specialists, the 4CCD method is a suitable alternative for the post-mortem evaluation of the fetal heart. 

The proposed 4CCD method has the merit to be easily performed following easily identifiable anatomic landmarks, does not require trained fetal pathology specialists, and can be performed even on extremely friable or small fetal hearts where the inflow–outflow method is difficult, if not impossible, to perform. 

In our study the prolonged formalin fixation time was required to allow for pm-MRI examination at 7T without tissue damage, especially for very small fetuses (15). Prolonged formalin fixation is known to cause an increased tissue consistency that, in the case of the 4CCD method, would facilitate proper cardiac sectioning (19). In our study, we had only fetuses with a low degree of maceration. We consider that the 4CCD method could be appliable even to severely macerated small fetal hearts that maintain soft consistency and increased friability even after prolonged formalin fixation using the adapted 4CCD protocol for fetal heart weights below 1 g. However, further studies are required to assess its accuracy considering the tissular changes that appear secondary to maceration that could influence the microscopic examination of the cardiac sections.

The main limitation of this study was represented by the small number of cases and the lack of normal values for the fetal cardiac structures according to the gestational age, which made the appreciation of slightly dilated or stenotic anatomic structures difficult since it was highly subjective. 

A limitation of the method is the fact that a good evaluation is highly dependent on correct macroscopic sectioning of the fetal heart. As observed previously, an oblique macroscopic sectioning can lead to incomplete visualization of several structures and to the possibility of either a misdiagnosis of lesions or overdiagnosis of normal structures as pathologic. Still, we observed that in all cases where correct sectioning of the fetal heart was achieved, all targeted anatomical structures were well visualized and properly evaluated morphologically. 

## 5. Conclusions

This study showed that the 4CCD method can be used as an alternative to the conventional inflow–outflow dissection method of fetal hearts with a median weight of 1 g. Compared to pm MRI, the 4CCD method of fetal hearts showed very good efficiency in identifying all the targeted anatomical structures. More reserved results were obtained for the evaluation of the malformative status, though most discrepancies were related to the subjective appreciation of diameters.

## Figures and Tables

**Figure 1 diagnostics-12-00223-f001:**
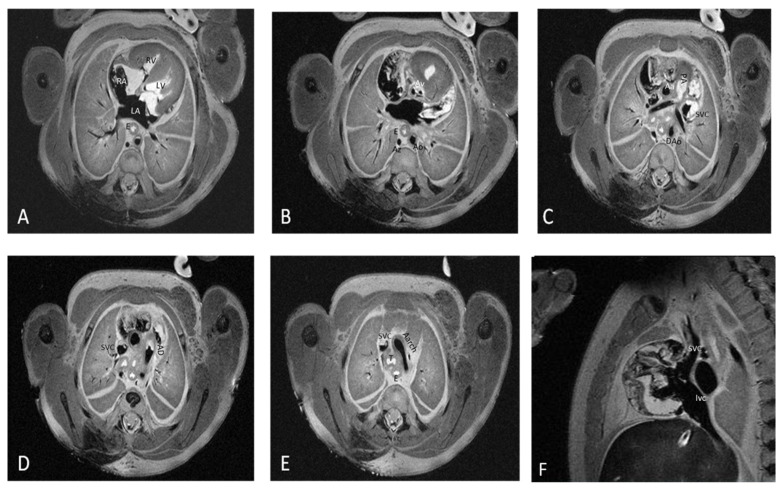
The fourteen cardiac items evaluated using 7T pm-MRI (T2, WI) depicted in a control case. Seventeen weeks of gestation-age fetus resulted from therapeutic interruption of pregnancy due to severe hydrocephaly with normal cardiac imaging evaluation and was confirmed by conventional stereomicroscopic dissection. Transvers section through the mediastinum depicting: (**A**) four chamber view with interatrial and interventricular septum, atrioventricular valves, left atrium (LA), right atrium (RA), left ventricle (LV), right ventricle (RV) and two pulmonary arteries, and esophagus (**E**); (**B**) aortic valve (Av), esophagus (**E**), aorta (Ao), and azygos vein (Az); (**C**) pulmonary artery (PA), ascending aorta (Aa), descending aorta (Dao), and superior vena cava (SVC); (**D**) arterial duct (AD); (**E**) aortic arch (Aarch) and trachea (T); and (**F**) sagittal section through the mediastinum depicting inferior vena cava (IVC) and SVC.

**Figure 2 diagnostics-12-00223-f002:**
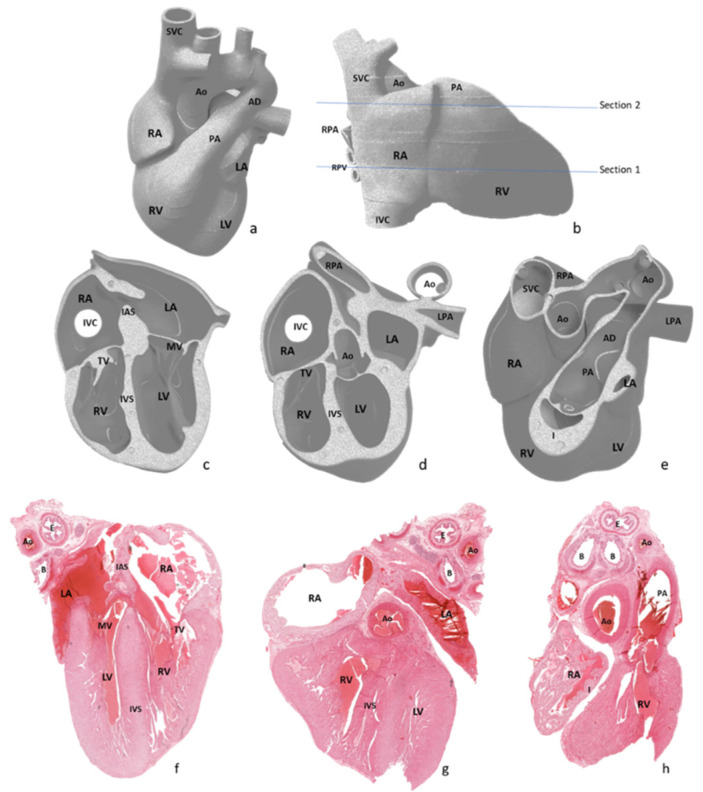
Graphical representation of macroscopic view of the four-chamber view cardiac dissection method and the anatomical structures evaluated. Abbreviations: RA—right atrium, RV—right ventricle, LA—left atrium, LV—left ventricle, Ao—aorta, PA—pulmonary artery, AD—arterial duct, IAS—interatrial septum, IVS—interventricular septum, IVC—inferior vena cava, SVC—superior vena cava, RPA—right pulmonary artery, RPV—right pulmonary veins, LPA—left pulmonary artery, TV—tricuspid valve, MV—mitral valve, I—infundibulum, E—esophagus, and B—bronchi. (a–e) graphic representation of heart sections used for the 4CCD method. (**f**) The 4CCD heart section and hematoxylin-eosin stain, 0.4×. (**g**,**h**) Heart section and hematoxylin-eosin stain, 0.5×.

**Figure 3 diagnostics-12-00223-f003:**
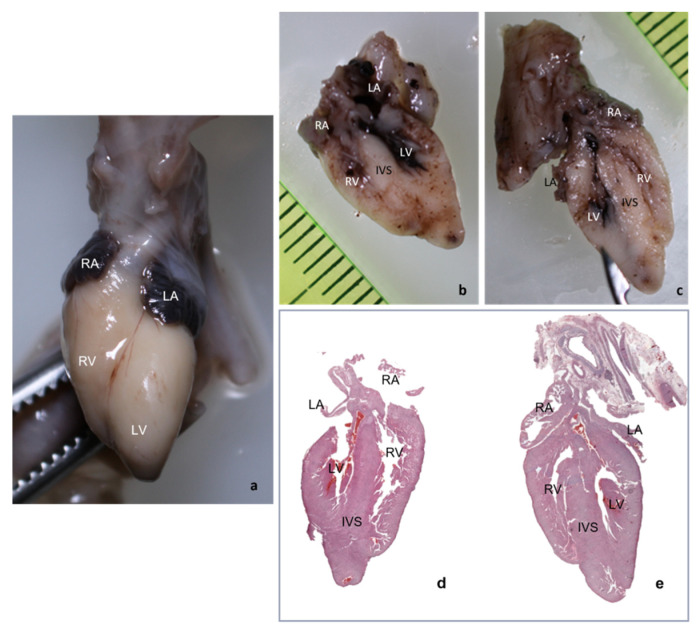
Mirrored view of microscopic slides as compared to macroscopy. (**a**) Macroscopic anterior view; (**b**,**c**) macroscopic inferior section; and (**d**,**e**) microscopic panoramic view, intermediate section. Hematoxylin eosine (HE). Abbreviations: RA—right atrium, RV—right ventricle, LA—left atrium, LV—left ventricle, Ao—aorta, and IVS—interventricular septum. (**d**,**e**) The 4CCD heart section and hematoxylin-eosin stain, 0.4×.

**Figure 4 diagnostics-12-00223-f004:**
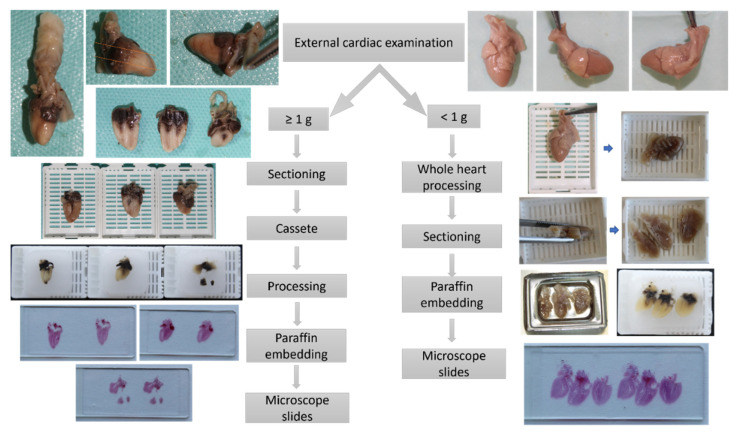
Workflow of proposed pathologic four-chamber examination method depending on the fetal heart weight.

**Figure 5 diagnostics-12-00223-f005:**
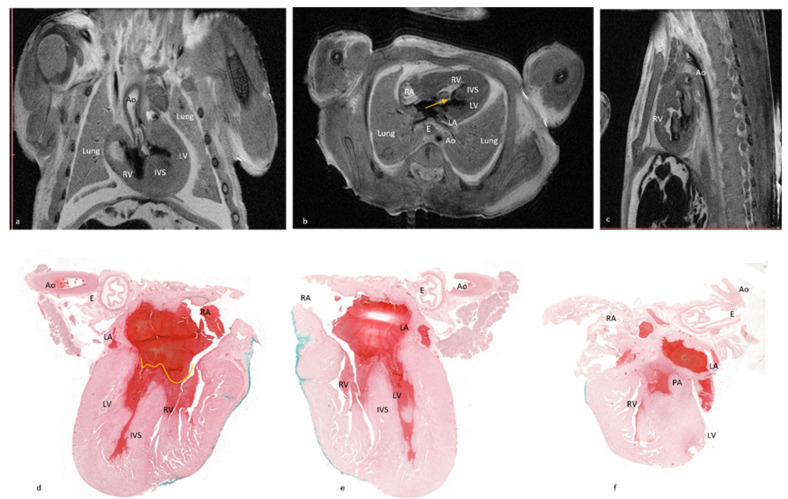
Presenting comparative images of the pm-MRI and microscopy of case No. 11. Note the small-sized left ventricle (**a**,**b**) and overriding aorta (**a**). (**c**) Sagittal section through mediastinum. Complete atrioventricular canal was observed on both examination methods. On the pm-MRI image (**b**), yellow arrow marks the ventricular septal defect and atrial septal defect. On the microscopic image (**d**), the yellow line highlights the atrioventricular valve passing over the ventricular septal defect. Interatrial septum is also absent. (**d**) The 4CCD heart section and hematoxylin-eosin stain, 0.4×. (**e**,**f**) The 4CCD heart section and hematoxylin-eosin stain, 0.5×.

**Figure 6 diagnostics-12-00223-f006:**
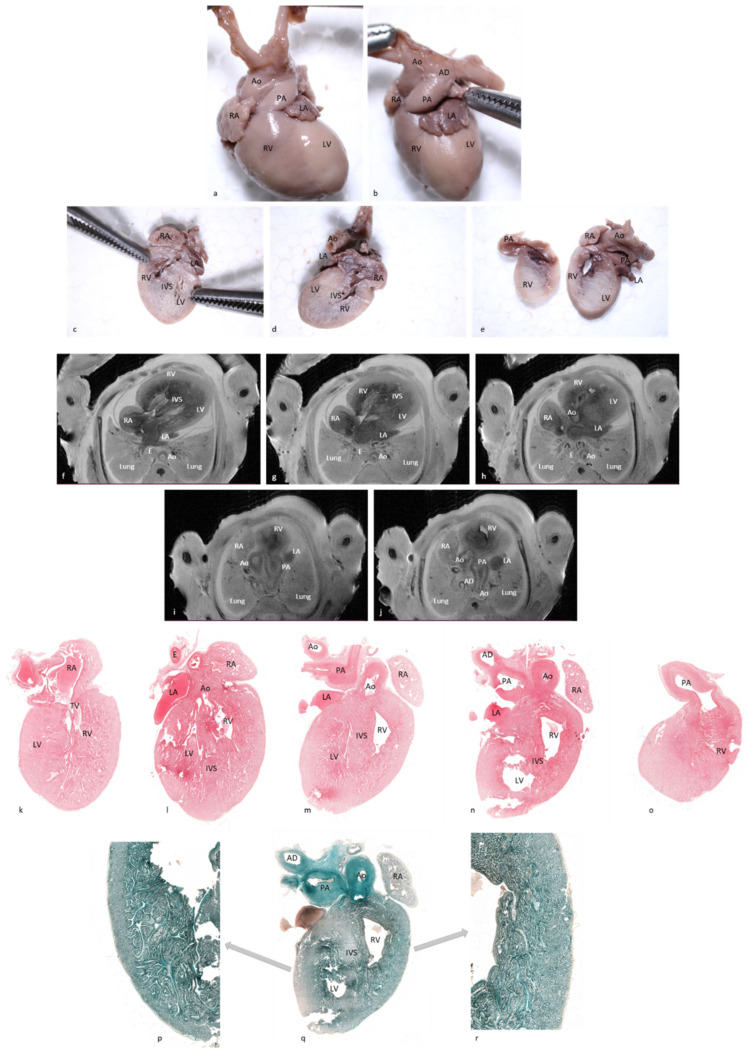
Comparative examination of pm-MRI at 7T (T2, WI) and four-chamber cardiac dissection of a 16-week gestation-age male with 18 Trisomy resulted from a therapeutic interruption of pregnancy (case 12 in the text). (**a**,**b**) Macroscopic anterior and antero-lateral view of the fetal heart. Thickening of great vessels and biventricular fibroelastosis. Thickened endocardium visible on pm-MRI images (**f**,**g**,**h**,**i**,**j**) but also on microscopic images ((**p**)—Masson’s trichrome stain, 1.4×; (**q**)—Masson’s trichrome stain, 0.7×; and (**r**)—Masson’s trichrome stain, 2.7×). Additionally, the thickening of great vessels, which can be observed on pm-MRI images, microscopy slides ((**k**)—hematoxylin-eosin stain, 0.6×; (**l**)—hematoxylin-eosin stain, 0.6×; (**m**)—hematoxylin-eosin stain, 0.5×; (**n**)—hematoxylin-eosin stain, 0.7×; and (**o**)—hematoxylin-eosin stain, 0.8×), and also on macroscopic sections (**c**,**d**,**e**). Abbreviations: RA—right atrium, RV—right ventricle, LA—left atrium, LV—left ventricle, Ao—aorta, PA—pulmonary artery, AD—arterial duct, IAS—interatrial septum, IVS—interventricular septum, IVC—inferior vena cava, SVC—superior vena cava, RPA—right pulmonary artery, RPV—right pulmonary veins, LPA—left pulmonary artery, TV—tricuspid valve, MV—mitral valve, I—infundibulum, E—esophagus, and B—bronchi.

**Table 1 diagnostics-12-00223-t001:** Specific characteristics and diagnostics for all cases included in our study.

Case Nr.	Maternal Age (Years)/Gestational Age (WG)/Fetal Sex * (M, F, and A)	Fetal Weight (g)/Fetal Heart Weight (g)	Pm MRI Examination –Cardiac Lesions	Pathologic 4CCD Method –Cardiac Lesions	Diagnosis Agreement Pm-MRI/4CCD –Cardiac Lesions	Associated Anomalies Detected at Conventional Autopsy
1	32 years/18 WG/M	192 g/1.6 g	Coarctation of the aorta, preductal form. Aortic arch anomaly.	Coarctation of the aorta, preductal form. Aortic arch anomaly.	CA	Facial dysmorphism. Trisomy 21.
2	31 years/18 WG/M	68 g/1 g	Ventricular septal defect.	Ventricular septal defect.	CA	Facial dysmorphism. Cystic hygroma. Unilateral (left) cleft lip and palate.
3	29 years/16 WG/M	100 g/0.23 g	Aortic stenosis. Left ventricle increased in size and cardiac apex was occupied only by the left ventricle.	Aortic stenosis. Left ventricle increased in size and cardiac apex was occupied only by the left ventricle.	CA	Facial dysmorphism. Laparoschisis. Nuchal edema. Bowel malrotation.
4	23 years/19 WG/M	364 g/1 g	Hypoplastic left heart syndrome. Complete absence of interatrial septum. Ventricular septal defect. Dilated arterial duct.	Hypoplastic left heart syndrome. Complete absence of interatrial septum. Ventricular septal defect. Dilated arterial duct.	CA	Facial dysmorphism. Polysplenia. Trisomy 21. Generalized edema.
5	18 years/17 WG/F	150 g/1 g	Without structural anomalies.	Without structural anomalies.	CA	Occipital meningoencephalocele. Hydrocephaly.
6	37 years/17 WG/M	214 g/1.95 g	Without structural anomalies.	Without structural anomalies.	CA	Facial dysmorphism. Dandy–Walker anomaly. Limb anomaly. Hypospadias. Cleft palate.
7	25 years/17 WG/F	98 g/1 g	Pulmonary artery stenosis. Arterial duct stenosis.	Without structural anomalies.	IA	Anencephaly. Pulmonary hypoplasia.
8	27 years/13 WG/A	23 g/1.02 g	Aortic stenosis. Left ventricle hypertrophy. Dilated left atrium.	Aortic stenosis. Ventricular septal defect.	IA	Facial dysmorphism. Hydrocephaly. Omphalocele. Bowel malrotation. Limb anomalies. Triploidy.
9	25 years/13 WG/F	33 g/0.12 g	Aortic stenosis. Biventricular hypertrophy.	Aortic stenosis. Hypoplastic left ventricle.	IA	Facial dysmorphism. Cystic hygroma. Bowel malrotation.
10	28 years/13 WG/M	42 g, 0.26 g	Without structural anomalies.	Without structural anomalies.	CA	Holoprosencephaly. Cleft lip and palate. Facial dysmorphism.
11	36 years/16 WG/M	80 g, 1 g	Complete atrioventricular canal. Hypoplastic left ventricle. Overriding aorta. Arterial duct agenesis.	Complete atrioventricular canal.	IA	Rachischisis. Cystic hygroma. Omphalocele. Facial dysmorphism.
12			Severe biventricular endocardial fibroelastosis. Less severe endocardial fibroelastosis of atriums. Thickening of great vessels and arterial duct walls. Bilateral ventricular hypertrophy. Ventricular septal defect.	Severe biventricular endocardial fibroelastosis. Less severe endocardial fibroelastosis of atriums. Thickening of great vessels walls and arterial duct wall. Aortic valve stenosis. Tricuspid valve stenosis.	IA	Facial dysmorphism. Pulmonary hypoplasia. Bowel malrotation. Unilateral renal agenesis. Thymus hypoplasia. Trisomy 18.

* M—male, F—female, and A—ambiguous. CA—complete agreement and IA—incomplete agreement.

**Table 2 diagnostics-12-00223-t002:** Showing the analysis of the effectiveness of the four-chamber cardiac dissection method in diagnosing fetal cardiac anomalies compared to pm-MRI for each examined structure.

Anatomic Structure	Sensitivity (CI 95%)	Specificity (CI 95%)	PPV (CI 95%)	NPV (CI 95%)	Accuracy (CI 95%)	Cohen’s Kappa
RA	50 (95% CI 1.26–98.74)	100 (95% CI 69.15–100)	100 (95% CI 2.5–100)	90.91 (95% CI 58.72–99.77)	91.67 (95% CI 61.52–99.79)	0.63
LA	33.33 (95% CI 0.84–90.57)	100 (95% CI 66.37–100)	100 (95% CI 2.5–100)	81.82 (95% CI 48.22–97.72)	83.33 (95% CI 51.59–97.91)	0.43
IAS	100 (95% CI 29.24–100)	88.89 (95% CI 51.75–99.72)	75 (95% CI 19.41–99.37)	100 (95% CI 63.06–100)	91.67 (95% CI 61.52–99.79)	0.8
TV	100 (95% CI 15.81–100)	100 (95% CI 69.15–100)	100 (95% CI 15.81–100)	100 (95% CI 69.15–100)	100 (95% CI 73.54–100)	1
MV	100 (95% CI 2.5–100)	90.91 (95% CI 58.72–99.77)	50 (95% CI 1.26–98.74)	100 (95% CI 69.15–100)	91.67 (95% CI 61.52–99.79)	0.63
RV	50 (95% CI 1.26–98.74)	100 (95% CI 69.15–100)	100 (95% CI 2.5–100)	90.91 (95% CI 58.72–99.77)	91.67 (95% CI 61.52–99.79)	0.63
LV	75 (95% CI 19.41–99.37)	87.5 (95% CI 47.35–99.68)	75 (95% CI 19.41–99.37)	87.5 (95% CI 47.35–99.68)	83.33 (95% CI 51.59–97.91)	0.63
IVS	66.67% (95% CI, 9.43–99.16)	88.89% (95% CI, 51.75–99.72)	66.67% (95% CI, 9.43–99.16)	88.89% (95% CI, 51.75–99.72)	83.33% (95% CI, 51.59–97.91)	0.56
PA	66.67 (95% CI 9.43–99.16)	100 (95% CI 66.37–100)	100 (95% CI 15.81–100)	90 (95% CI 55.5–99.75)	91.67 (95% CI 61.52–99.79)	0.75
Ao	85.71 (95% CI 42.13–99.64)	100 (95% CI 47.82–100)	100 (95% CI 54.07–100)	83.33 (95% CI 35.88–99.58)	91.67 (95% CI 61.52–99.79)	0.83
AD	25 (95% CI 0.63–80.59)	100 (95% CI 63.06–100)	100 (95% CI 2.5–100)	72.73 (95% CI 39.03–93.98)	75 (95% CI 42.81–94.51)	0.31
CV	100%	100%	NaN	NaN	NaN	1
PV	100%	100%	NaN	NaN	NaN	1
I	100%	100%	NaN	NaN	NaN	1

Abbreviations: RA—right atrium, LA—left atrium, IAS—interatrial septum, TV—tricuspid valve, MV—mitral valve, RV—right ventricle, LV—left ventricle, IVS—interventricular septum, PA—pulmonary artery, Ao—aorta, AD—arterial duct, CV—caval veins, PV—pulmonary veins, and I—infundibulum.

## Data Availability

All relevant data are within the manuscript.

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
