# Peer review of "The Evaluation of the Four-Chamber Cardiac Dissection Method of the Fetal Heart as an Alternative to Conventional Inflow–Outflow Dissection in Small Gestational-Age Fetuses"

_diagnostics, 2022, doi:10.3390/diagnostics12010223_

Round 1
Reviewer 1 Report
Albu et al. investigated the diagnostic capacity of four-chamber cardiac dissection techique in examining hearts of twelve fetuses with gestational age between 13 and 19 weeks using post-mortem MRI at 7T (pm-23 MRI) as standard. I think it is an interesting and methodologically correct study that investigates a complex, very specialistic (and rarely discussed) topic. Time since death should be made explicit for each case. Line 221: add mean values instead of median values (or both mean and median values). Magnification and stain should be added to each figure legend referring to microscopic images. Table 1 contains technical data that should be added as supplementary material. English language is proper but some sentences should be made clearer (e.g., line 69).
Author Response
We want to thank the reviewers and editor for their time and consideration of our manuscript and for their recommendations that we founded to be constructive and helpful. We have done a revision of our manuscript following point-by-point the reviewer’s recommendations. In the following section we will detail all the aspects that have been modified in the manuscript:
Response to Reviewer 1:
Time since death should be made explicit for each case.
All cases included in our study, resulted from therapeutic termination of pregnancy for fetal structural anomalies diagnosed during ultrasound examination. The procedure was carried out according to the internal protocol of 1st Clinic of Obstetrics and Gynecology Cluj-Napoca department at the time of recruitment, using prostaglandins and oxytocin. Mifepristone, known to induce fetal death in utero was not used for the cases included in our study, therefore, the retention in utero of abortion was not a problem. None of the pregnancies referred for therapeutic termination of pregnancy were for antepartum fetal death. The gestational age was calculated from the date of the last menstrual cycle. After delivery each case was stored 48 hours to 1 week in 10% formalin solution until examination. All cases had a low degree of maceration at the time of examination . We have completed the information in the Manuscript section 2.1. General considerations.
Line 221: add mean values instead of median values (or both mean and median values).
We completed the result section with the required values.
Magnification and stain should be added to each figure legend referring to microscopic images.
We added magnification and stain for each figure containing microscopic images, according to the values provided by the digital slide viewer software used for image acquisition.
Table 1 contains technical data that should be added as supplementary material.
We deleted the table from the manuscript test and added it as Supplemental material: Table I. We also changed the numbers of remaining tables and modified the text referring to the mentioned table, as follows: “The sequence parameters are listed in supplemental material Table I.”
English language is proper but some sentences should be made clearer (e.g., line 69).
We reformulated the indicated sentence as follows: “To our best knowledge the utility of the four-chamber view method for fetal cardiac examination was not evaluated until now. Although it was described previously as a dissection technique, it’s use was considered to bring less information compared to other dissection techniques.”
We also reformulated previous paragraph as follows: “Due to these facts, we considered introducing another cardiac dissection method, respectively the four-chamber cardiac dissection (4CCD) method, as an examination tool for small gestational age fetuses. “
We consulted a native English speaker, in order to correct grammatical errors and improve readability of our study (File attach: Manuscript).
Reviewer 2 Report
The aim was to evaluate if the 4CCD method can visualize standard cardiac structures and to verify the diagnostic concordance between pm-MRI cardiac examination and the 4CCD method. As a result, the 4CCD was able to show accuracy compared to pm- MRI.
These results were informative and valuable for readers.
Specific comments
Abstract and results
“The 4CCD was able to identify all cardiac anatomic structures as compared to pm- MRI, demonstrating a sensibility of 95.8% [95% CI, 94.5-95.8] and Specificity of 100% [95% CI, 32.3-100]. The overall accuracy was 95.8% [95% CI, 93.4-95.8]. The 4CCD method was able to detect cardiac anomalies with overall accuracy of 91% [95% CI, 85.8-94.2], sensibility of 67.6% [95% CI, 54.5-75.3] and specificity of 97% [95% CI, 93.7-99].“
These sentences were complicated.
What did 95.5% of the overall accuracy stand for?
What did 91% of cardiac anomalies with overall accuracy stand for?
Please clarify and explain what the difference between two numbers.
In addition, what did these numbers come from?
Introduction
The authors mentioned the inflow-outflow method.
Did the authors compare the results between 4CCD and inflow-outflow method?
If yes, please show the results.
In discussion, it may be better to discuss how 4CCD has merit comparing to inflow-outflow method.
Author Response
We want to thank the reviewers and editor for their time and consideration of our manuscript and for their recommendations that we founded to be constructive and helpful. We have done a revision of our manuscript following point-by-point the reviewer’s recommendations. In the following section we will detail all the aspects that have been modified in the manuscript:
Response to Reviewer 2
Abstract and results
“The 4CCD was able to identify all cardiac anatomic structures as compared to pm- MRI, demonstrating a sensibility of 95.8% [95% CI, 94.5-95.8] and Specificity of 100% [95% CI, 32.3-100]. The overall accuracy was 95.8% [95% CI, 93.4-95.8]. The 4CCD method was able to detect cardiac anomalies with overall accuracy of 91% [95% CI, 85.8-94.2], sensibility of 67.6% [95% CI, 54.5-75.3] and specificity of 97% [95% CI, 93.7-99].“
These sentences were complicated.
What did 95.5% of the overall accuracy stand for?
What did 91% of cardiac anomalies with overall accuracy stand for?
Please clarify and explain what the difference between two numbers.
In addition, what did these numbers come from?
We reformulated the mentioned paragraphs to allow better understanding of data, as follows:
Abstract: “The overall accuracy in identifying cardiac anatomic structures was 95.8% [95% CI, 93.4-95.8]. Also, the 4CCD method was able to detect cardiac anomalies with an overall diagnostic accuracy of 91% [95% CI, 85.8-94.2],…”
Results: “When we assessed the malformative status of the examined cardiac structures, the overall diagnostic accuracy of 4CCD compared to pm-MRI at 7T (used as golden standard) in detecting cardiac anomalies of the examined structures, was 91% [95% CI, 85.8-94.2],…”
To clarify the difference between numbers and how they were obtained we decided to reformulate several paragraphs in our manuscript, as follows:
Abstract: we reformulated the aim of the study to emphasize the fact that we are evaluating the examination efficiency of the proposed dissection method, which refers not only to its ability to observe specific anatomic structures, but also to evaluate their malformative status. “The aim of this study is to evaluate the efficiency of the 4CCD method in the examination of small fetal hearts using post-mortem MRI at 7T as standard. “
Introduction: we reformulated the aim of the study to make clear the two steps further mentioned in Results part. “The aim was to evaluate if the 4CCD method can visualize standard cardiac structures compared to pm-MRI examination at 7T.
Considering that 4CCD method can examine only few transversal sections of the fetal heart, a second objective was to verify the ability of 4CCD method to assess the malformative status of the evaluated cardiac structures compared to pm-MRI cardiac examination at 7T.”
Material and methods (Statistical analysis): we reformulated the sentence referring to our data base to emphasize that we are referring to two data sets (first referring to identification of anatomic structures, and second referring to the malformative status of the anatomical structures). “The information was centralized in a database. For each cardiac structure, it’s presence and assessed malformative status were noted separately. All information was analyzed by a statistician without medical knowledge.”
Discussions: we added a paragraph to further clarify the differences between results. “As expected, there was a difference between efficiency markers 4CCD method having a higher accuracy in detecting anatomic cardiac structures (95.8% [95% CI, 93.4-95.8]), and a lower accuracy of 91% [95% CI, 85.8-94.2] in assessing their malformative status. As opposed to pm-MRI, 4CCD method allows examination of only few cardiac sections, and some small cardiac anomalies could be missed at microscopic examination. Also as further presented, an incorrectly performed section could lead to misdiagnosis or incomplete diagnosis.”
Introduction
The authors mentioned the inflow-outflow method.
Did the authors compare the results between 4CCD and inflow-outflow method?
If yes, please show the results.
In discussion, it may be better to discuss how 4CCD has merit comparing to inflow-outflow method.
The inflow-outflow dissection method is the elective dissection technique for fetal hearts because it allows the visualization and complete assessment of all cardiac structures. If performed correctly by a specialized pathologist, it can evidentiate any structural anomaly of the fetal heart, and it should be used whenever possible. The proposed dissection method is addressed to those cases where cardiac examination of small fetuses using conventional method is not possible due to the lack of proper equipment or trained specialists.
We decided to use for the first time pm-MRI, a noninvasive method, as standard considering previously published data that observed good diagnostic accuracy of pm-MRI for fetal heart examination, compared to post-mortem conventional cardiac examination method.
We addressed this subject in the following paragraph added in Discussions:
“As compared to standard inflow-outflow dissection method, considered to be the gold standard for fetal cardiac examination, as it can evidentiate any structural cardiac anomaly, the 4CCD method has an accuracy of 91% [95% CI, 85.8-94.2] in detecting cardiac anomalies. We consider that conventional dissection should be used whenever possible. However, when the examination of small fetal hearts by conventional dissection is difficult due to maceration or increased tissular friability, or impossible due to the absence of specific medical instruments and trained specialists, the 4CCD method is a suitable alternative for the post mortem evaluation of the fetal heart.”
As recommended, we also added in Discussions following paragraphs to better emphasize the merit of 4CCD comparing to inflow-outflow method in selected cases, as for its possible further use in macerated severely macerated fetuses where conventional dissection is sometimes challenging.
“The proposed 4CCD method has the merit to be easily performed following easily identifiable anatomic landmarks, does not require trained fetal pathology specialists and can be performed even on extremely friable or small fetal hearts, where inflow-outflow technique is difficult if not impossible to perform.
In our study the prolonged formalin fixation time was required to allow pm-MRI examination at 7T without tissue damage especially for very small fetuses (15). Prolonged formalin fixation is known to cause an increased tissue consistency that in the case of 4CCD method would facilitate proper cardiac sectioning (19). In our study we had only fetuses with low degree of maceration. We consider that 4CCD method could be appliable even to severely macerated small fetal hearts that maintain soft consistency and increased friability even after prolonged formalin fixation using the adapted 4CCD protocol for fetal heart weights below 1g. However further studies are required to assess its accuracy, considering the tissular changes that appear secondary to maceration that could influence microscopic examination of the cardiac sections.”
Round 2
Reviewer 2 Report
The manuscript was well revised.